# Effects of Soil Bund and Stone-Faced Soil Bund on Soil Physicochemical Properties and Crop Yield Under Rain-Fed Conditions of Northwest Ethiopia

**Mulat Guadie [1], Eyayu Molla [2], Mulatie Mekonnen [3],* and Artemi Cerdà [4]**

[1]  Mertule Mariam College of Natural Resources Management, Enbse Sarmider,
    Mertule Mariam P.O. Box 012, Ethiopia; mulatguadie@gmail.com
[2]  College of Agriculture and Environmental Sciences, Department of Natural Resource Management,
    Bahir Dar University, Bahir Dar P.O. Box 5501, Ethiopia; eyayuelza@yahoo.com
[3]  College of Agriculture and Environmental Sciences, Department of Natural Resource Management and
    Geospatial Data and Technology Center, Bahir Dar University, Bahir Dar P.O. Box 1188, Ethiopia
[4]  Soil Erosion and Degradation Research Group, Department of Geography, Valencia University,
    Blasco Ibàñez, 28, 46010 Valencia, Spain; artemio.cerda@uv.es
*   Correspondence: mulatiemekonneng@gmail.com

**Abstract:** Research-based evidence on the effects of soil and water conservation practices (SWCPs) on soil physicochemical properties and crop yield is vital either to adopt the practices or design alternative land management strategies. Thus, this study was conducted to evaluate the effects of about 10-year-old soil bund (SB) and stone-faced soil bund (SFSB) structures on selected soil physicochemical properties, slope gradient, barley grain yield, and yield components in the Lole watershed, in the northwest highlands of Ethiopia. The experiment consisted of three treatments: (i) fields treated with SB, (ii) fields treated with SFSB, and (iii) fields without conservation practices (control) with three replications at three slope classes. A total of 27 composite soil samples from 0 to 20 cm depth and barley grain yield samples from 27 locations were collected. The soil samples were analyzed for bulk density, soil texture, porosity, soil reaction, cation exchange capacity, organic carbon, total nitrogen, available phosphorous, and potassium. Barley grain yield was analyzed using different agronomic parameters. The result indicated that SB and SFSB positively influenced the physicochemical properties of soils and barley grain yield. The interslope gradient between the successive SBs and SFSBs was reducing. Moreover, the untreated fields showed significantly lower barley grain yield, plant height, and straw biomass. Hence, SB and SFSB practices were found to be effective in changing slope gradient, improving soil fertility, and increasing crop yield. Therefore, this finding is vital to create awareness and convince farmers to construct SWCPs on their farmlands for sustainable land management.

**Keywords:** soil bund; stone-faced soil bund; interslope gradient; soil quality; grain yield; sustainable development goals

## 1. Introduction

Land and the fertility of its soil are critical natural capitals essential for sustainably ensuring food security, renewable energy, and water availability while eradicating rural poverty, conserving terrestrial biodiversity, and building the resilience of agricultural systems to climatic shocks [1–4]. However, land is also becoming vulnerable to soil fertility declination and associated changes in physical and chemical properties [5,6]. Due to these and other significant factors, soil need to be protected in a sustainable manner.

Soil conservation is a need to reduce soil fertility depletion and achieve sustainable land management, which is non-negotiable in developing countries where agriculture is the main source of labor and supplies the food for a growing population. Soil conservation should not reduce the agriculture production; on the contrary, it should increase the production, but conservation and productivity do not always come together. Agriculture is the mainstay of the Ethiopian economy, contributing approximately 41% of the gross domestic product (GDP), 84% of the total exports, and 80% of the employment [1,2,7,8]. However, soil erosion seriously limits agricultural productivity [9,10], declining soil fertility [11,12] and significantly reducing crop yields [13,14]. According to [15], annual soil loss was estimated at 1.5 billion metric tons, of which 50% occurs on croplands, which is highly pronounced in the Ethiopian highlands [16,17].

As part of the country, the northwestern highlands of Ethiopia, which form the Amhara National Regional State (ANRS), is seriously affected by soil erosion and soil fertility depletion, registering 58% of the total soil loss of the country [18,19]. Erosive tropical rains, steep slopes, extensive deforestation for fuelwood collection, the expansion of cultivation into steep land areas, overgrazing, long periods of maladapted agricultural practices, and high population pressure are important causes of such high rates of soil erosion [20–22].

Many studies in Ethiopia confirmed the positive impacts of soil and water conservation practices (SWCPs) on soil physicochemical properties and crop yields. For instance, soil conservation practices tested in Simada district, northwest Ethiopia, significantly improved the soil physicochemical properties [23], in which the clay content of the soils showed a significant difference among soil bund, stone bund, and control fields. A significantly higher amount of clay content was also found in treated fields i.e., grassed bunds (33%), soil bunds (28%), and stone bunds (29%) compared with the untreated fields (24%). Similarly, significantly lower mean bulk density was found in fields treated with SWCPs than the untreated fields in Adaa Berga district, western Ethiopia [24]. Other studies conducted in Ethiopia and other countries also verified the positive impacts of SWCPs on soil physicochemical properties and crop yields [11,12,17,20,25–31]. Moreover, [17] evaluated the 20-year-old SWCPs on slope gradient, and found a 2.7% slope reduction on average because of the trapped sediment.

The SWCPs improve the soil physicochemical properties (soil quality) [23,24], increase crop production and productivity [6,23,32], and reduce land degradation neutrality challenges [33,34], and thereby will help to attain the 2030 United Nations Sustainable Development Goals (UN-SDGs) such as ending poverty (Goal No. 1), ending hunger (Goal No. 2), good health and well-being (Goal No. 3), sustainable economic growth (Goal No. 8), sustainable production (Goal No. 12), climate change mitigation (Goal No. 13), and halting and reversing land degradation (Goal No. 15).

In Ethiopia, although many studies confirmed the positive impacts of SWCPs on soil physicochemical properties and crop yields, farmers frequently destruct SWCPs constructed on their fields, claiming that the practices didn't show a positive impact/effect other than occupying their farmlands. Such claims need investigation and measured data to design alternative land management strategies. Moreover, understanding how SWCPs reduce soil erosion, the loss of soil nutrients, and its impact on crop yield, is important to show and convince farmers of the effectiveness of such practices. Therefore, this study investigated the impacts of two commonly implemented structures—soil bund (SB) and stone-faced soil bund (SFSB)—on slope gradient, soil properties, barely grain yield, and yield components in the Lole watershed, in the northwest highlands of Ethiopia. The Lole watershed is well-known for its inappropriate land use, high population pressure, overgrazing, and erosive tropical rains, which are causing severe soil erosion [35,36]. To heal the causes of such soil erosion and alleviate the problem, governmental and non-governmental organizations have extensively implemented SB and SFSB structures. Moreover, no quantitative evidence has been reported on the impacts of SB and SFSB on soil fertility improvement and crop productivity in the watershed [35,36].

## 2. Materials and Methods

### 2.1. Study Area

The study was conducted in the Lole watershed in the northwest highlands of Ethiopia (UTM 1201658–1204571 N; 416690–418563 E; Adindan_UTM_Zone_37 N; Figure 1). It covers an area of 336 ha with an elevation range of 2436 m at the outlet to 2840 m above sea level at its highest point on the watershed divide. The mean annual minimum and maximum temperatures of the site are 10 °C and 22.5 °C, respectively, and the average annual rainfall is 1050 mm [36]. About 80–90% of the rainfall falls in the main rainy season (June–August), but is preceded and followed by one month of low and dispersed rains. Land use/cover of the study area includes cultivated/crop land (288.6 ha; 86%), shrubs and plantation (10.8 ha; 3%), grazing land (32.9 ha; 10%), and settlements (3.7 ha; 1%) [36].

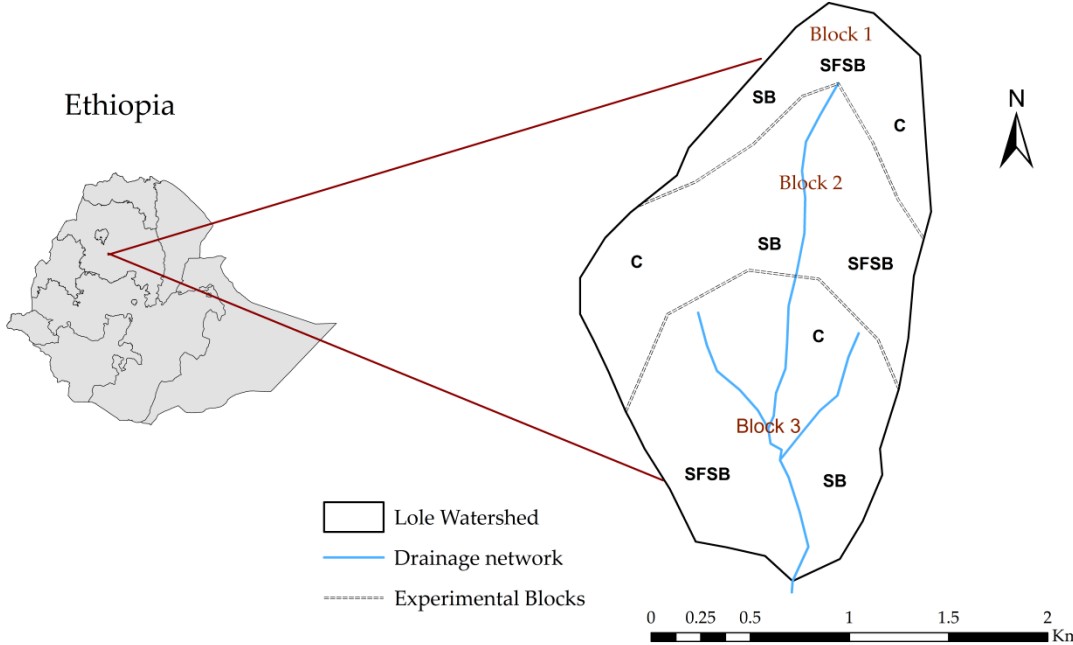

**Figure 1.** Location map of Lole watershed showing experimental blocks; block 1, block 2 and block 3, and the experimental treatments: soil bund (SB), stone-faced soil bund (SFSB), and control (C), in the northwest highlands of Ethiopia.

The watershed is characterized by mixed farming that is crop and livestock production. The crops grown are Teff (*Eragrostis tef*), wheat (*Triticum aestivum*), barley (*Hordeum vulgare*), and chickpea (*Cicer arietinum*). The major livestock is cattle, sheep, goat, and equine. The dominant soil types are Vertisols (15%), Lithic Leptosols (35%), and Nitosols (50%) [35,36]. The Vertisols were dominantly found in the valley bottoms (lower slopes), where there were no SWCPs.

Different soil and water conservation practices (SWCPs) have been implemented in Lole watershed since 2008 by governmental and non-governmental organizations with the participation of farmers through mass mobilization [35,36]. Soil bund (SB) and stone-faced soil bund (SFSB) were the dominant practices, which was the main reason to select this watershed for this investigation. About 50% of the watershed was treated with SB, about 35% was treated with SFSB, and the remaining 15% was untreated [36]. This untreated part (15%) is mainly dominated by Vertisols.

SB and SFSB are among the physical SWCPs that could reduce the velocity of runoff and consequently soil erosion, and the steady decline of crop yields [17,18,35]. They are impermeable structures that are intended to retain runoff from rainfall in the moisture stress areas and to drain the excess runoff in the moisture excess areas. Through their water retention effect, SB and SFSB may allow some crop yield, even in drought years [35]. SB is constructed from soil alone, and SFSB is constructed

from both stone and soil. In SFSB, the soil is reinforced by stone on one or both sides, and it has the same objectives as the SB.

## 2.2. Experimental Design

The experiment was conducted in the 2017/18 rainy season on farmers' fields under natural conditions. First, the study watershed was divided into three similar blocks; block 1, block 2, and block 3 (Figure 1) based on slope, soil type, and land use/cover to minimize variability. Then, the experimental fields were grouped randomly within the block, and separate randomizations were made for each block. Slope of the watershed ranged from 0–20% and three slope classes < 9% (lower), 10–14% (middle), and >15% (upper) were considered during block classification [37,38]. The experiment was designed in three treatments: (i) fields treated with soil bund (SB), (ii) fields treated with stone-faced soil bund (SFSB), and (iii) fields without soil and water conservation practices (control, C) in three replications with randomized complete block design. This experimental design is selected since it considers the natural farmers' fields as experimental plots. Moreover, it is cost-effective and better at representing the watershed scale environmental variables than the plot experiment that most researchers used for the last decades [17,18,39].

## 2.3. Soil Sampling

Soil samples were collected from each experimental field; fields treated with SB, fields treated with SFSB, and fields without SWCP (C) at the depth of 0–20 cm, assuming that the deposited sediment depth due to the implemented SWCPs will not exceed this depth [18,22,38]. Soil samples were collected about 1.5 m upslope of the investigated practices (Figure 2, below), assuming that sediment will be trapped and deposited up to this distance upslope of the structures [17,30,31]. From each experimental field, five soil samples were collected and mixed thoroughly, and a single composite sample was taken for analysis. A total of 27 composite soil samples were collected for soil physicochemical analysis, and 27 undisturbed soil samples were collected for bulk density determination.

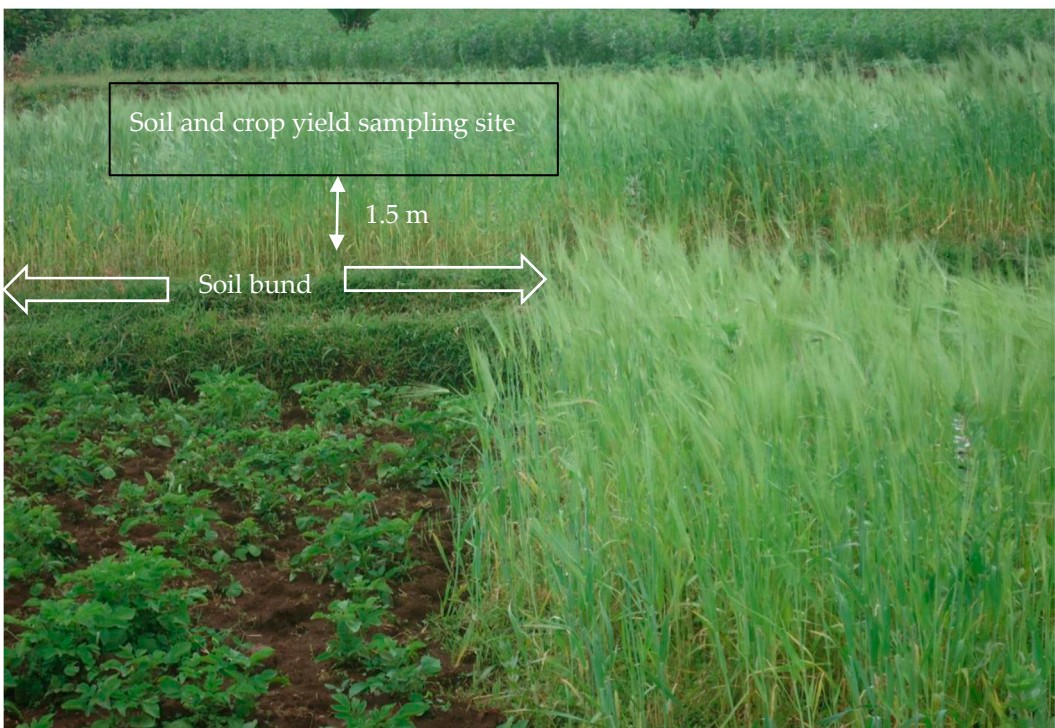

**Figure 2.** One of the soil and barley grain yield sampling fields treated with soil bund in Lole watershed. Sampling was done approximately 1.5 m upslope of the soil bund.

### 2.4. Crop Yield and Yield Components Sampling

Barley (*Hordeum vulgare*), the dominant crop in the study area, was used as a test crop. Barley grain yield and straw biomass samples were collected from the same sites where the soil samples were taken. A quadrant with 2 m × 2 m area within the sediment deposition zone, up upslope from the constructed SWCPs (Figure 2), was used to collect data on barley grain yield, plant height, and straw biomass for the treated fields. Yield samples were also collected from untreated fields as a control treatment. Plant height was measured from five representative plants randomly selected from each quadrant a week before harvesting, and their average was taken. The crop was harvested when it was ready for harvest, and grain yield was separated from the straw by hand and weighed. The straw biomass was determined by taking the sundry weight of barley collected from each quadrant.

### 2.5. Laboratory Analysis

The disturbed composite soil samples collected from the experimental fields were air-dried, mixed well, and passed through a 2-mm sieve for soil physicochemical analysis. The analysis was carried out in the soil testing laboratory of Adet Agricultural Research Center. Soil texture was analyzed following the Bouyoucos hydrometer method [40]. Bulk density was estimated from undisturbed soil samples using the core method, as described in [41]. The total porosity of the soil was derived from bulk and particle densities using the following Equation [41]:

$$f(\%) = \left(1 - \frac{Bd}{Pd}\right) \times 100 \tag{1}$$

where *f* is total porosity (%), *Bd* means soil bulk density, and *Pd* means soil particle density with an average value of 2.65 g cm$^{-1}$.

Soil pH was measured potentiometrically in the supernatant suspension of a 1:2.5 soil-to-water ratio mixture using a digital pH meter [42]. Soil organic carbon (SOC) was determined by the wet oxidation method [43], and the total nitrogen (N) content of the soil was determined using wet digestion, by the Kjeldahl method [44]. The available phosphorus content of the soil was determined using the Olsen extraction method [45], available potassium was determined by extracting the soil sample with Morgan's solution, and K in the extract was measured by flame photometer [46]. The cation exchange capacity (CEC) was determined using an ammonium acetate saturation method at pH 7.0 [47].

### 2.6. Statistical Analysis

The influence of independent variables (soil bund, stone-faced soil bund, and slope) on the dependent variables, soil properties, and crop yield were analyzed using Excel and IBM SPSS statistics 22 software. Mean separation was made using the least significant difference (LSD). Also, Pearson correlation analyses were carried out to see the relationships within the different soil parameters. Repeated ANOVA with soil sample characteristics, crop grain yield, and yield components were run to evaluate the difference in soil physicochemical properties and crop yield between the treated and untreated fields and among slope classes.

## 3. Results and Discussion

### 3.1. Effects of Soil Bund and Stone-Faced Soil Bund on Soil Physical Properties

Soil texture: Clay, silt, and sand fractions were significantly affected ($p \leq 0.05$) by soil bund (SB), stone-faced soil bund (SFSB), and slope gradients (Table 1). The overall mean sand fraction was found to be high in the upper (>15%) and low in the lower (<9%) slope positions. However, the silt and clay fractions were higher in the lower (<9%) slope positions. In general, sand content increases as slope gradient increases, and clay and silt content decreases as slope gradient increases. This could be due to the selective removal and transport of fine soil particles such as clay and silt by water erosion to

the lower slope, leaving the coarser materials onsite in the upper slope positions. The result agreed with the reports of [11] that showed an increase in sand and decline in silt and clay contents with an increase in slope gradient in the Weday watershed, eastern Ethiopia. According to [48], sands are easily detachable but difficult to transport; in contrast, silt and clay are easily transportable although they are difficult to detach by runoff water.

**Table 1.** Effects of soil and water conservation practices (SWCPs) and slope on soil physical properties. SB: soil bund, SFSB: stone-faced soil bund.

| Soil Properties | Slope Class | SWCPs/Treatments | | | |
| --- | --- | --- | --- | --- | --- |
| | | Control | SB | SFSB | Overall Mean |
| Clay | Lower (<9%) | 38.67 ± 1.15 [a] | 45.33 ± 2.31 [b] | 45.33 ± 1.15 [b] | 43.11 ± 3.62 [A] |
| | Middle (10–14%) | 35.33 ± 4.62 [a] | 43.33 ± 1.15 [b] | 42.67 ± 3.05 [b] | 40.44 ± 4.70 [A] |
| | Upper (>15%) | 30.00 ± 2.00 [b] | 36.00 ± 3.46 [a] | 36.00 ± 3.46 [a] | 34.00 ± 4.00 [B] |
| | Overall | 34.66 ± 4.58 [b] | 41.56 ± 4.94 [a] | 41.33 ± 4.8 [a] | |
| | LSD 0.05 | | 2.7223 | | |
| Silt | Lower (<9%) | 40.67 ± 1.15 [a] | 37.33 ± 1.15 [b] | 36.67 ± 0.0 [b] | 38.22 ± 2.11 [A] |
| | Middle (10–14%) | 30.67 ± 3.05 [d] | 35.33 ± 1.15 [b] | 36.00 ± 2.00 [b] | 34.00 ± 3.16 [B] |
| | Upper (>15%) | 24.00 ± 1.15 [c] | 28.67 ± 2.31[d] | 28.00 ± 1.1 [d] | 26.89 ± 2.67 [C] |
| | Overall | 31.78 ± 7.51 [b] | 33.78 ± 4.17[a] | 33.56 ± 4.3 [a] | |
| | LSD 0.05 | | 1.7469 | | |
| Sand | Lower (<9%) | 20.67 ± 2.31 [c] | 17.33 ± 1.15 [b] | 18.00 ± 0.0 [b] | 18.67 ± 2.00 [C] |
| | Middle (10–14%) | 34.00 ± 2.00 [b] | 21.33 ± 1.15 [a] | 21.33 ± 2.3 [a] | 25.56 ± 6.54 [B] |
| | Upper (>15%) | 46.00 ± 0.00 [a] | 35.33 ± 1.15 [c] | 36.00 ± 3.4 [c] | 39.11 ± 5.49 [A] |
| | Overall | 33.56 ± 11.08 [a] | 24.66 ± 8.25 [b] | 25.11 ± 8.5 [b] | |
| | LSD 0.05 | | 1.8282 | | |
| BD (gm cm$^{-3}$) | Lower (<9%) | 0.98 ± 0.02 [b] | 0.78 ± 0.11 [c] | 0.83 ± 0.02 [c] | 0.86 ± 0.11 [B] |
| | Middle (10–14%) | 1.08 ± 0.09 [a] | 0.94 ± 0.09 [b] | 0.97 ± 0.07 [b] | 0.99 ± 0.09 [A] |
| | Upper (>15%) | 1.13 ± 0.11 [a] | 0.98 ± 0.07 [b] | 1.01 ± 0.03 [b] | 1.04 ± 0.07 [A] |
| | Overall | 1.06 ± 0.06 [a] | 0.90 ± 0.12 [b] | 0.94 ± 0.09 [b] | |
| | LSD 0.05 | | 0.0621 | | |
| Porosity (%) | Lower (<9%) | 62.87 ± 0.59 [a] | 70.39 ± 4.46 [b] | 68.81 ± 1.13 [b] | 67.36 ± 4.14 [A] |
| | Middle (10–14%) | 59.17 ± 0.36 [b] | 64.54 ± 3.47 [a] | 63.24 ± 2.71 [a] | 62.32 ± 3.28 [B] |
| | Upper (>15%) | 57.45 ± 0.43 [b] | 62.99 ± 2.74 [a] | 61.85 ± 1.24 [a] | 60.76 ± 3.00 [B] |
| | Overall | 59.83 ± 2.43 [b] | 65.97 ± 4.61 [a] | 64.63 ± 3.57 [a] | |
| | LSD 0.05 | | 2.3431 | | |

Mean values followed by different small letters (a, b, c) along the same rows and capital letters (A, B, C) along the same column are significantly different at $p \leq 0.05$. LSD is least significant difference and BD is bulk density.

On the other hand, a statistically significant difference ($p \leq 0.05$) was found in clay, silt, and sand proportion between treated and untreated fields. The overall mean percentage of clay and silt content was significantly higher in the treated than the untreated fields, whereas the sand fraction was significantly lower in the treated than the untreated fields (Table 1). This might be due to the accumulation of fine-textured clay and silt fractions behind the constructed structures. The result concurs with the findings of [26] in Rwanda, [27] in southern Ethiopia, and [23] in northwest Ethiopia, in which higher clay and silt proportions were found in fields treated with SWCPs than the untreated fields.

Soil bulk density (BD): BD showed a statistically significant difference ($p \leq 0.05$) between the treated and untreated fields and among slope positions (Table 1). BD was found to be lower in fields treated with SB and SFSB than the control. Higher BD in the untreated fields could be associated with the absence of SWCPs that exposed the soil to erosion and consequently to the removal of organic carbon from the topsoil layer. This finding was in line with those of [24] and [5], which showed significantly lower BD values in the treated micro-watersheds than the untreated in Adaa Berga district, western Ethiopia, and Ambachia watershed, northern Ethiopia, respectively.

Similarly, BD showed a statistically significant variation ($p \leq 0.05$) at different slope positions. It was found to be lower in lower (<9% slope) than in the upper (>15% slope) positions. As slope gradient increases, BD increases, which could be associated with low soil organic matter content. [24]

reported lower BD in cultivated fields of lower slope positions than in the upper slope in Adaa Berga district, Western Shewa, Ethiopia. Other studies [12,49] in the Goromti watershed and in the Guto Gida District, Western Ethiopia, respectively, also reported the direct relationship of BD and slope gradient. This study also showed significant and negative correlation of BD with clay fraction (r = −0.76 **); significant and positive correlation with sand fraction (r = 0.73 **), and significant and negative correlation with organic matter (r = −0.70 **). The reason could be associated with variations in soil organic matter content, which has an inverse relationship with soil BD.

Soil porosity: Soil porosity showed a statistically significant difference ($p \leq 0.05$) at different slope positions (Table 1). In general, the values of soil porosity decrease as the slope gradient increases. The lowest soil porosity in the upper slope fields (>15%) might be due to the intensive cultivation and soil erosion, which reduces the soil organic matter content and total pore volume of the soil. The result agrees with those of [39], which reported lower total porosity in steep slope than in gentle slope fields as a result of high BD, low clay content, and low organic matter content in the Dawja watershed, northwest Ethiopia.

A statistically significant difference ($p \leq 0.05$) was found in soil porosity between the treated and untreated fields. Lower soil porosity (59.83 ± 2.43) was found in the untreated fields than the treated fields. The low soil porosity in the untreated fields might be due to the low organic matter content of the soil as a result of soil erosion that caused higher BD. This result agreed with the findings of [23] in northwest Ethiopia that low soil porosity in the untreated/control field was due to the removal of soil organic matter and exposure of the subsoil as a result of soil erosion. The total soil porosity showed a significant and positive correlation with organic matter OM (r = 0.70 **) and a significant and negative correlation with bulk density (r = −1.00 **; Table 6). The highest soil porosity was recorded in fields located in the lower slope position (<9%) having the highest clay content showing the positive effect of clay content on soil porosity. This result was in line with [40] in which the lowest total soil porosity (46.42%) was recorded in fields having steep slope, while the highest total soil porosity (50.10%) was recorded on fields having a gentle slope in the Dawja watershed, northwest Ethiopia. Soil texture, bulk density, and porosity didn't show a statistically significant difference ($p \leq 0.05$) between SB and SFSB, which might be due to the similarity in the age of the practices.

## 3.2. Effects of Soil Bund and Stone-Faced Soil Bund on Soil Chemical Properties

Soil Reaction (pH): Soil pH showed a statistically significant difference ($p \leq 0.05$) between the treated and untreated fields (Table 2). It was 6.51 ± 0.32 behind the SB, and 6.48 ± 0.26 behind the SFSB and 5.90 ± 0.48 in the control treatment. This might be due to the effect of soluble bases and organic matter removal through sheet erosion from the control fields due to the absence of SWCPs, as it was reported by [11] in the Weday watershed, eastern Ethiopia. Similarly, [28] indicated low pH values in the untreated fields due to the low base saturation percentage and low sediment organic matter (SOM) content and high pH value in the sediment accumulation zone behind the SWCPs of the treated fields in the Anjeni watershed, central highlands of Ethiopia. In general, as per the ratings of [50], the soil pH in the Lole watershed was slightly acidic (5.9–6.65), which is suitable for crop production, as most nutrients for field crops are available at pH values between 5.5–7.0 [33].

**Table 2.** Effects of SWCPs and slope on soil chemical properties. CEC: cation exchange capacity.

| Soil Properties | Slope Class | SWCPs/Treatments | | | |
| --- | --- | --- | --- | --- | --- |
| | | Control | SB | SFSB | Overall |
| PH | Lower (<9%) | 6.35 ± 0.07 [b] | 6.77 ± 0.36 [a] | 6.81 ± 0.12 [a] | 6.65 ± 0.29 [A] |
| | Middle (10–14%) | 6.06 ± 0.09 [a] | 6.41 ± 0.32 [b] | 6.27 ± 0.75 [b] | 6.25 ± 0.23 [B] |
| | Upper (>15%) | 5.30 ± 0.10 [c] | 6.35 ± 0.11 [b] | 6.36 ± 0.06 [b] | 6.01 ± 0.54 [C] |
| | Overall | 5.90 ± 0.48 [b] | 6.51 ± 0.32 [a] | 6.48 ± 0.26 [a] | |
| | LSD 0.05 | | 0.1769 | | |
| CEC (cmolkg$^{-1}$) | Lower <9% | 28.24 ± 4.69 [b] | 40.69 ± 8.94 [a] | 40.60 ± 3.07 [a] | 36.51 ± 8.14 [A] |
| | Middle (10–14%) | 21.47 ± 3.92 [d] | 28.40 ± 4.95 [b] | 38.53 ± 1.39 [a] | 29.47 ± 8.10 [B] |
| | Upper (>15%) | 18.88 ± 2.00 [c] | 21.10 ± 1.27 [c] | 30.40 ± 0.87 [b] | 23.46 ± 5.44 [C] |
| | Overall | 22.86 ± 5.28 [c] | 30.06 ± 10.00 [b] | 36.51 ± 4.98 [a] | |
| | LSD 0.05 | | 4.1688 | | |
| OC (%) | Lower (<9%) | 1.81 ± 0.30 [c] | 2.21 ± 0.31 [b] | 2.79 ± 0.07 [a] | 2.27 ± 0.48 [A] |
| | Middle (10–14%) | 1.58 ± 0.19 [d] | 1.82 ± 0.21 [d] | 2.42 ± 0.31 [b] | 1.94 ± 0.43 [B] |
| | Upper (>15%) | 0.92 ± 0.16 [b] | 1.25 ± 0.15 [c] | 1.39 ± 0.06 [c] | 1.18 ± 0.24 [C] |
| | Overall | 1.44 ± 0.45 [c] | 1.76 ± 0.47 [b] | 2.20 ± 0.65 [a] | |
| | LSD 0.05 | | 0.2157 | | |

Mean values followed by different small letters (a, b, c) along the same rows and capital letters (A, B, C) along the same column are significantly different at $p \leq 0.05$.

The variations in soil pH were also statistically significant ($p \leq 0.05$) in different slope positions. The overall mean value of soil pH was found to be low in the upper slope (>15%), and high in the lower (<9%) slope positions. As the slope gradient increases, soil pH decreases. This might be due to the influence of the slope gradient through its effect of facilitating soil erosion and the leaching of soluble base cations, which in turn increased the concentration of H$^+$ ion in the soil solution and reduced soil pH. This result agreed with the findings of [23] in Simada district, northwest Ethiopia. The difference in pH across the slope could also be associated with the distribution of SOM and CEC, as pH is positively and significantly correlated with SOM, CEC, and clay fraction (r = 0.66 **, r = 0.62 and r = 0.72, respectively).

Cation Exchange Capacity (CEC): CEC showed a statistically significant difference ($p \leq 0.05$) between the treated and untreated fields. Soils in the treated fields showed significantly higher CEC than the untreated fields. This finding implies that CEC was significantly influenced by the implementation of SWCPs, which might be due to the accumulation of SOM behind SWCPs. This was confirmed by the significant and positive correlation of SOM (r = 0.80 **) and clay content (r = 0.74 **) with CEC (Table 6). The result is in line with the reports of [23] and [11], in which higher mean CEC values were found in the treated than in the untreated fields in Adaa Berga district, central Ethiopia and the Weday watershed, eastern Ethiopia, respectively. Therefore, as per the ratings established by [51], the CEC value in the Lole watershed was found to be high, which might be linked to the higher content of the clay particles.

On the other hand, the CEC values showed a statistically significant difference ($p \leq 0.05$) at different slope gradients. It was found to be low in the upper slope positions (>15%), and high in the lower (<9%) slope positions. As the slope gradient increased, the CEC value decreased. This might be due to the removal of basic cations from the upper slope and accumulation in the lower slope positions. This result is in line with the findings of [23] and [11], in which higher CEC values in the lower slope were found than those in the upper slope positions in Adaa Berga district, central Ethiopia and the Weday watershed, eastern Ethiopia, respectively.

Organic Carbon (OC): Soil organic carbon (SOC) showed a statistically significant difference ($p < 0.05$) between the treated and untreated fields (Table 2), which might be associated with sediment accumulation due to SWCPs and crop residues in the treated fields. According to [42], due to SWCPs, high SOC (3.69%) was found in the treated Tsegur Kidanemihret micro-watershed compared with the untreated Tsegur Eyesus micro-watershed (2.24%), northwest Ethiopia. In general, as per the ratings

of [51], SOC content was found to be low in the Lole watershed, which might be due to intensive tillage, continuous cropping, and the removal of crop residues.

　　SOC also showed a statistically significant variation ($p < 0.05$) between the different slope positions (Table 2). Higher SOC was recorded at lower than higher slope gradients. SOC showed an inverse relationship with slope gradient; i.e., as slope gradient increases, SOC declines. This might be associated with the removal of organic matter from the higher slope areas and its subsequent deposition in the lower slope areas via water erosion. The result agrees with [30] and [13], who found that fertile soil deposition at a lower slope favored high crop biomass and residue, as well as SOC, in Mesobit-Gedba northern Ethiopia, and the Zikre watershed, northwest Ethiopia, respectively.

　　Total Nitrogen (TN): TN showed a statistically significant difference ($p \leq 0.05$) at different slope positions (Table 3). High TN was recorded in the lower slope than in the higher slope gradients. This might be due to the removal of organic matter from the steep slopes via soil erosion. Similar results were reported by [13,24] in the Zikre watershed, Adaa Berga district, and by [39] in the Dawja watershed, northwest Ethiopia.

**Table 3.** Effects of SWCP and slope on soil chemical properties.

| Soil Properties | Slope Class | SWCPs/Treatments | | | |
|---|---|---|---|---|---|
| | | Control | SB | SFSB | Overall |
| TN (%) | Lower (<9%) | 0.26 ± 0.06 [c] | 0.42 ± 0.03 [a] | 0.43 ± 0.03 [a] | 0.37 ± 0.10 [A] |
| | Middle (10–14%) | 0.17 ± 0.01 [a] | 0.30 ± 0.00 [b] | 0.32 ± 0.0 [b] | 0.26 ± 0.07 [B] |
| | Upper (>15%) | 0.14 ± 0.01 [b] | 0.21 ± 0.02 [c] | 0.29 ± 0.00 [d] | 0.22 ± 0.01 [C] |
| | Overall | 0.19 ± 0.06 [c] | 0.31 ± 0.09 [b] | 0.35 ± 0.07 [a] | |
| | LSD 0.05 | | 0.0279 | | |
| AV-p (mg kg⁻¹) | Lower (<9%) | 11.12 ± 2.90 [a] | 19.60 ± 0.74 [b] | 17.56 ± 1.40 [b] | 16.09 ± 3.94 [A] |
| | Middle (10–14%) | 7.17 ± 0.30 [b] | 10.96 ± 1.09 [a] | 14.66 ± 3.64 [d] | 10.93 ± 3.76 [B] |
| | Upper (>15%) | 5.54 ± 1.45 [c] | 6.62 ± 0.45 [c] | 6.64 ± 0.13 [c] | 6.27 ± 0.94 [C] |
| | Overall | 7.94 ± 2.63 [b] | 12.4 ± 5.76 [a] | 12.95 ± 5.27 [a] | |
| | LSD 0.05 | | 1.4833 | | |
| AV-K (mg kg⁻¹) | Lower (<9%) | 130.35 ± 8.11 [b] | 177.32 ± 9.72 [c] | 188.40 ± 10.64 [a] | 165.36 ± 27.94 [A] |
| | Middle (10–15%) | 94.03 ± 8.57 [a] | 137.74 ± 17.17 [b] | 132.57 ± 11.5 [b] | 121.45 ± 23.52 [B] |
| | Upper (>15%) | 83.94 ± 10.67 [a] | 99.89 ± 4.74 [d] | 110.48 ± 6.22 [c] | 98.10 ± 13.32 [C] |
| | Overall | 102.77 ± 22.58 [b] | 138.32 ± 35.04 [a] | 143.82 ± 35.77 [a] | |
| | LSD 0.05 | | 10.17 | | |

Mean values followed by different small letters (a, b, c) along the same rows and capital letters (A, B, C) along the same column are significantly different at $p \leq 0.05$.

　　Similarly, TN showed a statistically significant difference ($p \leq 0.05$) between the treated and untreated fields (Table 3). The treated fields showed higher TN values than the untreated fields, which could be associated with the implementation of SWCPs that maintain soil fertility by decreasing the removal of SOC and TN through soil erosion. This finding is in line with [31] and [29], who found that higher TN content was recorded in treated fields compared with untreated fields in southern Ethiopia and northwest Ethiopia, respectively. The Pearson correlation coefficient also revealed that TN significantly and positively correlated with SOM (r = 0.80 **) (Table 6). This is because SOM is the main source of TN. TN also correlated positively with SOC because of increased biomass production, litter quantity, and organic matter decomposition. In general, TN was low in the untreated fields and medium in the treated fields, as per the ratings suggested by [51], indicating that nitrogen is a limiting plant nutrient in the study area. This might be due to the limited use of nitrogen-containing inputs such as commercial fertilizer, plant residues, and animal manure.

　　Available Phosphorus (Av-P): Av-P showed a statistically significant difference between the treated and untreated fields (Table 3). Low Av-P from untreated fields was due to continuous cultivation without SWCPs, extractive crops biomass harvest, and soil erosion, as indicated by the findings of [11] and [29,31] in eastern and southern Ethiopia, respectively. Av-P also showed a positive and significant

relationship with SOM and TN (r = 0.85). The Av-P of the study watershed was medium based on the rating of [51].

Av-P significantly varied at different slope gradients ($p \leq 0.05$). Higher mean Av-P was recorded in the lower slope gradients than in the upper ones, which might be due to the washing out of topsoil and organic matter from the higher slope gradients and their subsequent accumulation at the lower gradient/deposition zone, which agrees with the findings of [11] in the Weday watershed, eastern Ethiopia, reference [40] in the Dawja watershed, northwest Ethiopia, and [31] in Wenago district, southern Ethiopia. In contrast to this result, [12] and [28] reported that the mean values of Av-P were not significant at different slope gradients in the Goromti watershed, western Ethiopia, and the Anjeni watershed, central highlands of Ethiopia, respectively.

Available Potassium (Av-K): Av-K showed a statistically significant difference between treated and untreated fields ($p < 0.05$; Table 3). Higher Av-K was recorded in the lower slope (<9%) than in the upper slope (>15%) positions due to the transportation of potassium by erosion from steep slope areas to gentle/low slope, as reported in the findings of [36] in Rwanda. In contrast to this, [12] in the Goromti watershed, western Ethiopia revealed that Av-K didn't show a significant difference ($p < 0.05$) at different slope gradients. In general, TN showed significant difference, but Av-p and AV-k didn't show a significant difference between fields treated with SB and SFSB.

### 3.3. Effects of Soil Bund and Stone-Faced Soil Bund on Slope Change

The average inter-terrace slope gradient showed a statistically significantly different ($p \leq 0.05$) between the treated and untreated fields. The average inter-terrace slope gradient in the treated fields was found to be low compared with the untreated fields (Table 4). However, the average inter-terrace slope gradient between soil bund (SB) and stone-faced soil bund (SFSB) didn't show a significant difference. The deposition of soil materials and debris on the upper position of SB and SFSB (usually called accumulation zone) causes the height of the bunds to upsurge year after year, thereby reducing the inter-terrace slope gradient between two successive structures, which is in line with the findings of [17] and [52].

**Table 4.** Effects of SWCPs on the inter-terrace slope.

| Treatment | Average Inter-Terrace Slope (%) | Bund Height (cm) |
|---|---|---|
| Soil bund | 7.16 [b] | 80.00 [a] |
| Stone-faced soil bund | 8.00 [b] | 67.00 [b] |
| Control (non-treated land) | 19.50 [a] | 00.00 [c] |
| LSD | 11.64 | 9.08 |
| CV | 49.67 | 9.28 |

Where CV is Coefficient of Variation. Mean values followed by different small letters (a, b, c) are significantly different at $p \leq 0.05$.

### 3.4. Effects of Soil Bund and Stone-Faced Soil Bund on Barley Grain Yield

A statistically significant difference ($p \leq 0.05$) was observed in plant height, grain yield, and straw biomass between treated and untreated fields and at different slope positions (Table 5). Barley height, grain yield, and straw biomass were found to be high on fields having <9% slope and low on fields having >15% slope because of topsoil and nutrient loss from upper slopes by erosion and deposition at the lower slope. Similarly, [32] reported a reduction in barley plant height, grain yield, and straw biomass as the slope gradient increases due to the loss of SOM, N, P, and K by erosion from steep slope areas (Table 6).

**Table 5.** Effects of SWCPs and slope on plant height, grain yield, and straw biomass.

| Yield & Yield Components | Slope Class | SWCPs/Treatments | | | |
|---|---|---|---|---|---|
| | | Control | SB | SFSB | Overall |
| Plant height (cm) | Lower (<9%) | 58.88 ± 5.85 [a] | 70.56 ± 5.36 [a] | 70.56 ± 6.74 [a] | 66.67 ± 7.82 [A] |
| | Middle (10–14%) | 48.89 ± 1.9 [b] | 65.00 ± 1.67 [b] | 65.89 ± 4.17 [a] | 59.93 ± 8.64 [B] |
| | Upper (>15%) | 52.78 ± 4.20b | 57.89 ± 3.42 [c] | 55.55 ± 6.94 [b] | 55.41 ± 4.93 [B] |
| | Overall | 53.52 ± 5.74 [b] | 64.48 ± 6.41 [a] | 64 ± 8.48 [a] | |
| | LSD 0.05 | | 4.7808 | | |
| Grain yield (q ha$^{-1}$) | Lower (<9%) | 16.38 ± 2.0 [c] | 24.44 ± 4.19 [d] | 25.55 ± 0.96 [d] | 22.13 ± 4.9 [A] |
| | Middle (10–14%) | 16.11 ± 0.96c | 23.88 ± 2.54 [d] | 22.22 ± 2.54 [b] | 20.74 ± 4.00 [A] |
| | Upper (>15%) | 12.77 ± 2.54 [a] | 18.33 ± 1.66 [b] | 20.00 ± 1.66 [b] | 17.04 ± 3.7[B] |
| | Overall | 15.10 ± 2.44 [b] | 22.22 ± 3.90 [a] | 22.59 ± 2.90 [a] | |
| | LSD 0.05 | | 2.3072 | | |
| Straw yield (q ha$^{-1}$) | Lower (<9%) | 42.5 ± 2.5 [a] | 63.33 ± 7.4 [c] | 68.33 ± 5.2 [b] | 58.06 ± 12.79 [A] |
| | Middle (10–14%) | 35 ± 5.00 [b] | 47.50 ± 2.5 [a] | 45.83 ± 1.44 [a] | 42.78 ± 6.55 [B] |
| | Upper (>15%) | 35 ± 5.00 [b] | 43.33 ± 2.89 [a] | 45.00 ± 5.00 [a] | 41.11 ± 6.01 [B] |
| | Overall | 37.50 ± 5.3 [b] | 51.39 ± 10.08 [a] | 53.06 ± 12.04 [a] | |
| | LSD 0.05 | | 4.47 | | |

Mean values followed by different small letters (a, b, c) along the same rows and capital letters (A, B, C) along the same column are significantly different at $p \leq 0.05$.

**Table 6.** Pearson's correlation matrix for soil physicochemical properties.

| | BD | Porosity | Ph | OC | OM | N |
|---|---|---|---|---|---|---|
| BD | 1 ** | | | | | |
| Porosity | −1 ** | 1 ** | | | | |
| Ph | −0.77 ** | 0.77 ** | 1 ** | | | |
| OC | −0.69 ** | 0.69 ** | 0.66 | 1 ** | | |
| OM | −0.69 ** | 0.69 ** | 0.66 ** | 1 ** | 1 ** | |
| N | −0.79 ** | 0.79 ** | 0.80 ** | 0.80 ** | 0.80 ** | 1 ** |
| P | −0.78 ** | 0.78 ** | 0.65 ** | 0.85 ** | 0.85 ** | 0.85 |
| CEC | −0.75 ** | 0.75 ** | 0.62 ** | 0.80 ** | 0.80 ** | 0.80 ** |
| K | −0.77 ** | 0.77 ** | 0.73 ** | 0.81 ** | 0.81 ** | 0.90 ** |
| Clay | −0.76 ** | 0.76 ** | 0.72 ** | 0.80 ** | 0.80 ** | 0.80 ** |
| Silt | −0.55 ** | 0.55 ** | 0.65 ** | 0.71 ** | 0.71 ** | 0.62 ** |
| Sand | 0.73 ** | −0.73 ** | −0.76 ** | −0.84 ** | −0.84 ** | −0.78 ** |

** Correlation is significant at the 0.01 level. Where OC is organic carbon and OM is organic matter.

A statistically higher overall means of plant height, grain yield, and straw biomass were recorded in the treated fields than the untreated ones (Table 5 and Figure 3). Plant height, grain yield, and straw biomass didn't show a statistically significant difference between fields treated with SB and SFSB, which might be due to similarity in the age of structures. Increased crop yield on the treated fields was due to reduced soil and nutrient losses, and improved soil fertility because of SWCPs. This result agreed with those of [6,23]. According to [32], higher crop yield was recorded in treated fields in Absela Kebele, northwest Ethiopia. The SB and SFSB structures take up the land that will be used for crop production, and farmers mostly plant grass on such practices, which can be used for livestock feed. Hence, future studies should consider this when comparing crop yield between the treated and untreated fields.

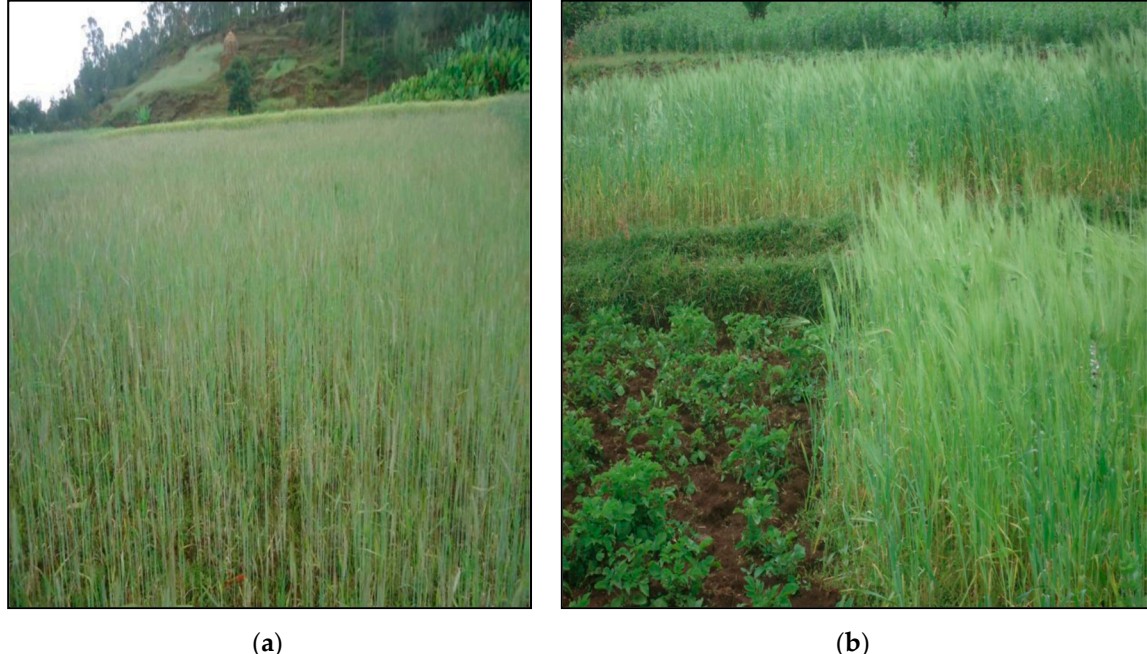

(**a**)　　　　　　　　　　　　　　　　　　　　　　　　　　(**b**)

**Figure 3.** Barley production. Poor performance in the untreated fields (**a**) and good performance in the treated fields (**b**).

## 4. Future Research Directions

Although physical, vegetative, and agronomic soil and water conservation practices (SWCPs) played a vital role in reducing soil erosion and enhancing sustainable land management individually, their integration/combination is more effective [17,22,53,54]. In Ethiopia, most of the farmers are claiming that the impact of SWCPs is long-term, not short-term; hence, age-based investigations on the effectiveness SWCPs is highly appreciated [55,56], which would help to achieve the sustainable development goals of the United Nations and the land degradation neutrality challenges [33,34]. One of the long-term objectives of SWCPs is to bring a change in landscape (slope gradient) through reducing slope length, which further reduces the speed and erosive capacity of runoff. Hence, the ability of the existing SWCPs in changing the length and angle of a farming land should be investigated, and practices that could bring such changes in a relatively short period of time should be identified and promoted. Also, the traditional or indigenous land management practices should be investigated and provided as alternatives for farmers. Few studies have been conducted to see the role of SWCPS, such as check dams and sediment storage dams, in stocking/sequestering carbon and sediment organic carbon [52]. Therefore, the importance of SWCPs in stocking carbon and reducing greenhouse gas concentration in the atmosphere has to be investigated.

## 5. Conclusions

Soil bund (SB) and stone-faced soil bund (SFSB) in the Lole watershed, in the northwest highlands of Ethiopia, positively influenced the physicochemical properties of soils and barley grain yield. Soil physicochemical properties (soil texture, bulk density, TN, Av-P, Av-K, SOM, pH, and CEC) showed a statistically significant difference between the treated and untreated fields, and at different slope gradients. Except for sand content, the other parameters (clay, silt, bulk density, TN, Av-P, Av-K, SOM, pH, and CEC) were found to be high in fields having lower slope gradients than in fields having higher slope gradients. Soils of the study area were found to be slightly acidic (5.9–6.65; pH), thus not affected by acidity-related problems, since most of the nutrients for field crops are available in this pH range.

Fields treated with SB and SFSB showed a statistically significant positive correlation in barely grain yield, plant height, and straw biomass compared with the untreated fields. This means that SB

and SFSB improved the fertility of the soil by reducing nutrient losses. Barley grain yield obtained on fields treated with SB (22.22 q ha$^{-1}$) was high compared to the control (15.10 q ha$^{-1}$). Barley grain yield obtained on fields treated with SFSB (22.59 q ha$^{-1}$) was also high compared to the control. This means that barley grain yield was 34% higher on fields treated with SB and SFSB. In general, the widely practiced 10-year-old SB and SFSB structures in the Lole watershed were found to be effective in changing slope gradient, improving soil fertility, and increasing crop yield.

**Author Contributions:** Conceptualization, M.G., E.M., and M.M.; methodology, M.G.; software, M.M.; validation, A.C., M.M., and E.M.; formal analysis, M.G.; investigation, M.M.; resources, A.C.; data curation, M.M.; writing—original draft preparation, M.G.; writing—review and editing, E.M.; visualization, M.M.; supervision, E.M.; project administration, M.M.; funding acquisition, A.C. All authors have read and agreed to the published version of the manuscript.

**Funding:** This research received no external funding.

**Acknowledgments:** The authors would like to thank Adet Agricultural Research Center for soil physicochemical property analysis. We also would like to thank the farmers and development agents for their assistance during the field work. We would also extend our thanks to Bahir Dar University College of Agriculture and Environmental Sciences (CAES), and the Geospatial Data and Technology Center (GDTC). The authors also acknowledge the inspiring ideas given by Saskia Keesstra, who knew Ethiopia very well. The comments of the anonymous reviewers were greatly appreciated.

**Conflicts of Interest:** The authors declare no conflict of interest.

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
