# Peer review of "Effects of Soil Bund and Stone-Faced Soil Bund on Soil Physicochemical Properties and Crop Yield Under Rain-Fed Conditions of Northwest Ethiopia"

_land, doi:10.3390/land9010013_

Round 1
Reviewer 1 Report
This article reports the effect of soil and water conservation practices like soil bund on soil basic physico-chemical properties. it is not clear how this study is different from existing similar articles. I would like the authors to highlight what is novel in this work. Otherwise, the manuscript is sound but needs English changes and proof reading.
Abstract: Could you specify which soil physical and chemical properties were measured?
Introduction: I would like to see the authors build more on sustainable soil and water conservation practices for slopes in the introduction. The knowledge gap is not clear. Why is this study important and how does it add to the literature is not very clear.
Materials and Methods: what is the proportion of each of the cultivated crops area-wise?
Results and Discussion: I see all tables and no figures for the data presented.
Author Response
Find attached herewith the reply to the reviewer.

Reviewer 2 Report
This research aimed at quantifying the effect of soil and water conservation practices (i.e. soil bund and stone-faced soil bund) on phsyico-chemical properties of soils as well as on crop yield. The research topic is interesting however the manuscript had several errors throughout. Please find my specific comments below:
The third paragraph of the introduction section mentioned that "Many studies in Ethiopia confirmed the positive impacts of SWCP......." However, only two other studies were cited in relation to this.
The font size was inconsistent in the introduction section.
Although this study is mainly focused on two SWCP (soil bund and stone-faced soil bund). However, information specifically related to these two practices is missing.
Caption for Fig. 1 needs formatting.
The way the experimental design described right now is confusing. In the experimental design section...at one place it is mentioned that "watershed is divided into 3 blocks based on similar physiographic conditions such as slope....and at another place it is mentioned that slope of watershed ranged from 0-20% and three slope classes <9% (lower), 10-14% (middle) and >15% (upper) were considered.........". Please rewrite for clarification.
How repeated ANOVA was used in statistical analysis is not clear?
All tables need reformatting. Row Space is inconsistent.
There are spelling and grammatical mistakes throughout the manuscript.
In the results and discussion section, comparisons were mainly made between upper and lower slope (middle slope) not described.
Similarly, comparisons were mainly made between control and treatment; the impact of individual treatments i.e. soil bund vs. stone-faced soil bund was missing.
Author Response
Find attached herewith the replay to the reviewer. Please see the corrections in the revised manuscript re-submitted.

Reviewer 3 Report
Arrange the order in which the papers are cited and keep the order of numbering.
Detailed notes can be found in the attached file pdf.

Author Response
Find attached herewith the reply to the reviewers. Please see all correction in the revised manuscript re-submitted.

Reviewer 4 Report
It is an interesting study and well written up. However, before it can be published some more work is needed. Please consider the following points.
This study was carried out in the semi-arid part of Ethiopia. That should be in the title. The effectiveness of these practices is quite different in the semi-arid area compared with the more humid Ethiopia highlands. It is therefore misleading not to emphasize this
Similarly, through the manuscript it is mentioned studies that support the result of the current study, the authors need also to mention the studies that does not agree with the findings
One of the weaknesses of the study is that the moisture contents were not measured in the year that the yields were measures. It is very likely the greater moisture contents in the fields with soil and water conservation practices caused the difference in crop yield. It is hard to believe that the differences in physical properties causes the great differences. The authors need to find other studies that can justify the differences in yield by soil physical and chemical properties alone. If this cannot be justified the word yield should be removed from the title.
Barley and barely are two completely different things. Check the manuscript.
The title has also a spelling error. It is rainfed not rained. Even better to replace this rained with semi-arid.
On page 2 “content was found in treated fields i.e., grassed bunds (32.48%), soil bund (28%) and stone bunds (28.98%) compared with the untreated fields (23.78%).” Please round of the integer numbers. The results are just not accurate justifying this many digits.
On page 2 the following is writes “The Lole watershed is well-known by the inappropriate land use, high population pressure, overgrazing, erosive tropical rains and long periods of inappropriate agricultural practices are using severe soil erosion [23, 24]. To treat such soil erosion causes and alleviate the problem,”
The assumption that farmer’s follow inappropriate practices is just not fair to the farmers. Usually farmers know what they are doing. What experts think is the best way just might be wrong” Moreover, the increases soil erosion is a direct consequence of the loss of organic matter. Please read the articles of Tebebu et al.. Please remove the statement about inappropriate use. It is an assumption by so-called experts, but it has never been proven.
In the material and methods (page 4) information on the climate and especially rainfall is missing. These are important to understand why the crop yields was so much different.
Page 5: since porosity and bulk density are related mathematically, they should be discussed in the same paragraph
Table 1 The numbers in Table 1 have too many digits. Please only report the digits that can be justified based on the accuracy of the measurements. This is likely not four digits
Same problem with tables 2 and 3
It would be better to present first the results o the chemical analysis and then explain the reason that there are differences. All the chemicals move downhill. No need to indicate for each chemical separately.
On page 11 and 12 it should be stressed that these are valid for the semi-arid regions for Ethiopia
On page 11 and 12 It is shown that the yields per unit plot area or indeed much greater for plots treated with soil and water conservation practices. The soil and water conservation practices take up land and the land taken out of production should be taken into account when presenting the results
Some more work needed but it is potentially interesting paper especially if the effect of amount of rainfall on the effectiveness of the practices is incorporated in the discussion of the results
Author Response
Find attached herewith the reply to the reviewer. please see all the correction in revised manuscript re-submitted.

Reviewer 5 Report
Dear Authors,
The manuscript (ID: land-591915) discusses the influence of conservation practices on soil physical and chemical properties. It is well written. However, there are a few things that needs to be addressed. With a Vertisol (50%), how did the researchers account for the natural cracks due to the shrink-swell clays? Furthermore, what kind of clays was dominant (smectite, kaolinite, montmorillonite??). This will provide better information on the CEC and nutrient availability of the soil. Also, there are some spelling and grammatical errors in the article (e.g. Page 5). Please correct those. Finally, there are no units for bulk density and porosity in table 1. These corrections should greatly improve the manuscript.
Author Response
Find attached herewith the reply to the reviewer. Please see all the correction in the revised manuscript re-submitted.

Round 2
Reviewer 1 Report
The authors have not addressed all the comment to the extent that was necessary.
Comment 1 and 3: The introduction still needs to be improved. The rationale for conducting this study needs to be strongly stated. The literature review is not sufficient.
Comment 2: Please include the physico-chemical properties in the abstract. Unless mentioned physico-chemical properties could be any soil property. The authors have analyzed the basic physico-chemical properties in this study which might not be sufficient to put it under the broad group 'physico-chemical properties'
Comment 4: An estimate of the crop area coverage will be very informative.
Comment 5: Please convert some tables to figures (illustrations).
Author Response
Uploaded here under
